# Venture Capital and Chinese Firms' Technological Innovation Capability: Effective Evaluation and Mechanism Verification

**Yuegang Song** [1], **Songlin Jin** [1] **and Zhenhui Li** [2,*]

1    School of Business, Henan Normal University, Xinxiang 453007, China
2    School of Economics and Management, Communication University of China, Beijing 100024, China
*    Correspondence: lipai@cuc.edu.cn

**Abstract:** Making the financial industry a solider mainstay of the real economy is of great concern for China in the midst of economic reform. For China, leveraging venture capital (VC) to enhance a firm's technological innovation capability (TIC) is an important means of actualising its innovation and development strategy, as well as a must-do to realise sustainable development. In this study, firms that went public from 2010 to 2020 on the A-stock market were used as samples to study the effects of VC on TIC and the relevant mechanism based on the difference-in-differences (DID) method. As research findings show, VC can improve TIC through the medium of the internal incentive and external constraint easing effects. The contributory role of VC in TIC varies with firm size, ownership, and industry type. A range of robustness tests, including the PSM, variable substitution, and instrumental variable methods, further strengthened the reliability of the conclusions. This study can enlighten policymakers on how to implement comprehensive resource factor market reform to build a favourable innovation environment that materialises the role of marketisation.

**Keywords:** venture capital; technological innovation capability; internal incentive effect; external constraint easing effect

## 1. Introduction

The current circumstances of China's economy, the world's largest developing country, is mired in an ebbing economy, a struggling extensive growth model, and an increasingly weakening conventional motive power [1]. In this context, it is important for China to boost technological innovation, invest more in basic research, drive technological breakthroughs, improve independent innovation capability, put in place a modern industry system, modernise the industry and supply chain, and attain its goal as a leading innovative economy by 2030. As the major champion of innovative activities, firms are supposed to solidify their foothold in innovation and concentrate on innovative factors to install a firm-led market-oriented technological innovation system built on equal stress of research and application [2,3]. The research and development (R&D) intensity of Chinese firms has increased from 1.2% in 2010 to 3.6% in 2020 (R&D intensity = R&D spending/net sales). Nevertheless, the R&D intensity of American firms is double than that of their Chinese counterparts (Data Sources: The 2021 EU Industrial R&D Investment Scoreboard). Compared with the U.S. and other developed economies, there are still many realistic challenges to address for China, such as a scarcity of talent and inadequate policy support, funds, and innovative resources.

Chinese firms are confronted with unprecedented challenges as trade friction escalates and the COVID-19 pandemic has spread worldwide. For start-up firms short of money, venture capital (VC), with its capital appreciation and innovation-inclined value-added effects, solves externality, information asymmetry, and the moral hazard of technological innovation financing, providing an important avenue for technological innovation capability (TIC) improvement and mattering a great deal to corporate sustainability [4,5]. In recent years, the Chinese government has rolled out a series of policies to incentivise VC

to leverage the niches in information asymmetry, financing, governance, and resource networks to help innovation-thirst small- and medium-sized enterprises (SMEs) out of straits and strengthen TIC [6]. Owing to government policies, VC in China has flourished exponentially in recent years, topping $130.6B and accounting for 21.03% of the global total (Data Sources: 2022 China Venture Capital Market Scale and Investment Industry Survey). Therefore, how to create the best VC to strengthen TIC is vital to the development of firms.

Innovation is a highly uncertain activity subject to multiple factors as the key driver of a firm's prosperity [7]. VC is a new financing model in which firms, particularly global Chinese Internet giants, take advantage of renewing a wave of innovation and entrepreneurship efforts. Although Chinese firms seem to have made use of VC to consolidate TIC, the U.S. government's suppression of ZTE and Huawei in the form of sanctions have made society more aware of the technological innovation potential of Chinese firms. Does VC contribute to TIC while China implements its national innovation initiative? Is it an innovation capability that draws VC or vice versa? How does VC affect TIC? Systematically answering the above questions cannot only update VC investment theories and corporate TIC but also improve company development and ongoing economic growth.

Compared with the extant study, we have the following potential marginal contributions:

First, in terms of theoretical mechanism, the extant study focuses on analysing VC's internal governance mechanisms for TIC, such as innovation spending, value-added services, and monitoring and control. No studies have dealt with the relationship between VC and TIC from the perspective of external mechanisms. By exploring the internal incentive and external constraint easing effects, we add to the study on the mechanism through which VC affects TIC and make the conclusion more cogent.

Second, in terms of research methods, the extant study makes analyses based simply on VC entry and overlooks the endogeneity problem arising from pre-entry differences. We combined the difference-in-differences (DID) model with the propensity score matching method to conduct a robustness test, which effectively averts interference factors and affirms the relationship between VC and TIC.

Third, from a research perspective, we dissect the impact and theoretical mechanism of TIC from the perspective of VC, thereby providing new empirical evidence of TIC improvement amid escalating trade friction and global propagation of the pandemic.

The remainder of this paper is organised as follows: Section 2 provides a discussion of the existing study, and Section 3 discusses the theoretical mechanism and research hypotheses. Section 4 discusses the model, variables, and data. Section 5 elaborates on the empirical results and analyses, and Section 6 concludes the study.

## 2. Literature Review and Research Innovation

Academia is a deepening study on VC, as this emerging business model has gone from strength to strength worldwide in recent years. According to Bottazzi et al. [8], although current studies on this subject concentrates mainly on whether VC affects the successful listing of start-up companies, academia rarely investigates the effects of VC on listed companies. This study intends to deepen and expand research on the effects of VC firms' TIC on listed companies. Extant studies have focused on analysing TIC-affecting factors, the relationship between VC and the firm, and that between VC and TIC.

### 2.1. TIC-Affecting Factors

TIC is crucial for strengthening the core competitive advantage of a firm, region, and country [9]. Innovation activities are characterised by enormous investments, a long cycle, output uncertainty, and difficulty in measuring value [10]. Scholars have studied TIC-affecting factors, such as capital structure, financial market development, financing models, resource integration, and corporate governance. In a questionnaire-based investigation of Mexican SMEs, Cuevas-Vargas et al. [11] used a quantitative method and cross-sectional design based on partial least squares structural equation modelling (PLS-SEM) and found that capital structure has a significant, either positive or negative, effect on TIC.

Irwan et al. [12] believed that market mechanisms, such as market creation, regulation, stabilisation, and legalisation, intensify the nonlinearity of TIC. For instance, the development of credit and stock markets is not instrumental in innovation, unless at a certain threshold. Wang et al. [13] identified the differential effect of various financial models on wind power industry policy and TIC, whereas internal financing inhibits the role of industry policy on TIC, and external financing does the reverse. Based on the upper echelon theory, Shouming et al. [14] discovered that in the technology, media, and telecom (TMT) sector, the impact of executive team attention to innovation (TMTAI) on TIC activities is regulated by the characteristics of corporate governance.

### 2.2. Effects of VC on a Firm

VC first emerged in the 1930s and has since come through four periods: germination, rapid development, decline, and regulation. Academia examines the relationship between VC and the firm primarily from business operations. Hellmann and Puri [15] gathered data on Silicon Valley start-up companies to investigate how VC affects internal organisation professionalisation and found that it is of help for HR development. Lockett [16] studied the mechanism of VC for firms and found that VC can improve corporate governance structure by providing value-added activities. By classifying VC activities, Kerr et al. [17] testified that angel investing has a more remarkable impact on improving the management and TIC of the investee. Jin et al. [18] built a multiple regression model and established that VC can achieve more value-added activities through the TIC medium. In a study on public companies traded in the secondary market, Xu et al. [19] discovered that VC can substantially improve IPO performance. Hu and Zhou [20] studied companies that went public on the Shanghai and Shenzhen Stock Exchanges from 2007 to 2016 and found that VC enhances financing efficiency by lowering financing costs. Based on the certification surveillance and moral hazard hypotheses, Li and Yan [21] found that VC is effective in inhibiting internal control deficiency, an action that increases with the VC shareholder equity ratio and decreases with ownership concentration. Nevertheless, the effects of VC on firms are not always negative. In a study conducted by Wang and Wang [22], not all forms of VC do good for firm productivity. According to Guo et al. [23], rather than bringing out the best in information superiority, VC causes adverse selection by strengthening the positive correlation between investment in R&D and IPO underpricing.

### 2.3. Effects of VC on TIC

Chinese and foreign scholars are currently divided into studies on VC and TIC. First, VC was deemed to be of help in improving TIC in much of the studies. In a comparison between firms with VC and those without VC, Sandro et al. [24] found that VC improves firm profitability as well as TIC. Cailou and Dehai [25] used propensity score matching (PSM) and the Poisson model to study listed energy companies and identified the contributory role of VC in TIC. Second, some researchers raised queries. For instance, Engel and Keilbach [26] held that VC increases innovation inertia because, with investment, a firm will be weary with TIC enhancement efforts. Third, some scholars hold that the relationship between VC and TIC is more than a simple linear one. Based on the 2004–2013 pre-IPO data on SMEs and secondary market-listed companies, Wen and Feng [27] applied the propensity score matching-difference-in-differences (PSM-DID) method and found that the effects of VC on TIC decrease before a rise, an indication of a U-shaped curve with its trough occurring in the IPO year or previous year.

Currently, it is time to review current studies on VC and TIC. There are deficiencies of the existing studies. First, in terms of theory, there is a lack of studies on VC and enterprise TIC within the same theoretical framework, particularly on the relationship between the two based on mathematical models. Second, in terms of empirical evidence, there is a lack of studies on the mechanism and heterogeneity of VC in technological innovation at the micro level. Most studies focus only on the average effect of VC and do not delve into the impact on heterogeneous firms. Third, in terms of sample selection, it is difficult

to provide convincing answers to the above questions because of the lack of typicality and representativeness of the study samples. Finally, in terms of conclusion, reviewing existing studies on VC and corporate TICs, many scholars have tested the effects of VC on corporate TICs from the perspectives of corporate behaviour, causality, and opportunity cost using different samples and have found positive and negative correlations between the two and nonlinearity. Therefore, this study selects 3993 listed companies from 2010 to 2020, and studies the mechanism of VC on the TIC of enterprises at the micro level, which compensates for the insufficiency of current studies and provides Chinese enterprises growth and capital market development based on theoretical support and decision-making.

## 3. Theoretical Mechanism and Research Hypothesis

Based on Melitz's heterogeneity firm model [28], we introduce two variables, TIC and VC decision-making, to explore the effects of VC on TIC and relevant theoretical mechanism. ① The internal incentive effect. In order to improve innovation, a TIC-favourable environment needs to be created within the enterprise, which not only means orderly management, but also represents sufficient resources. Although VC cannot directly improve the TIC of enterprises after entering, providing more sufficient funds and scientific management can promote enterprises to increase investments in R&D and executive compensation, and stimulate the potential of technological innovation of enterprises. In this study, (>1) stands for the innovation cost per unit when the enterprise makes factor input decisions after the entry of VC. The larger the value, the more variable costs that the enterprise needs to invest in innovation, and the greater the difficulty of innovation. ② The external constraint easing effect. Among the external constraints faced by enterprises, financing constraints have the greatest impact on enterprises, and this impact has a lot to do with the entry of VC. The reason is that the certification function of VC institutions can help enterprises to obtain financing more easily and relieve external financing environment of enterprises. This stands for the financing constraint, that is, the more alleviation of VC on corporate financing constraints, the less difficult to innovate.

### 3.1. Preferences and Demand

We assume that the consumer utility function is expressed in terms of the constant elasticity of substitution (CES):

$$U = \left[ \int_{i \in I} q_i^{\frac{\sigma-1}{\sigma}} di \right]^{\frac{\sigma}{\sigma-1}} \tag{1}$$

where $i$ represents the type of differentiated products; $q_i$ represents the consumer demand for differentiated products; and $\sigma(\sigma > 1)$ represents the elasticity of substitution between differentiated products. From the utility maximization principle, the consumer demand function for differentiated products $i$ can be deduced as:

$$q_i = \frac{pi^{-\sigma}}{P^{1-\sigma}} R \tag{2}$$

where $p_i$ is the price of differentiated products $i$ and $P$ is the price index of differentiated products, which is given by $P = \left( \int_{i \in I} p_i^{1-\sigma} di \right)^{\frac{1}{1-\sigma}}$; Total social income is $R = PQ = \int_{i \in I} r(i) di$.

### 3.2. Technological Innovation Spending Choice

Assuming that, according to Melitz [28], the enterprise obtains productivity $\varphi$ randomly after entering, and the specific form of the productivity distribution function is $S(\varphi) = \varphi^{\theta}, \phi \in [0,1], \theta > \sigma - 1$, where the smaller $\varphi$ is, the higher the enterprise productivity is. Enterprises that enter the industry and remain in the industry first decide to invest in innovation $\gamma_i$. Referring to the treatment method of Maria et al. [29], it is assumed that the success of innovation can improve the productivity of enterprises to $\varphi/\gamma_i$. The innovation cost of a firm is directly related to the initial productivity level, so $\varphi$ is used here

as the innovation cost per firm, indicating that the higher the initial productivity, the lower the innovation cost. After the VC enters, the innovation cost per unit when the enterprise makes the decision of factor input is $\varepsilon = a\varphi$, $a \in (0,1)$, which means that the innovation cost of enterprise unit decreases after the entry of VC.

*3.3. Enterprise Production Behavior*

Enterprises produce intermediate products by themselves and take innovation as the intermediate product in the production process. The production of intermediate products requires labor factor l, which is used in production and innovation activities. Set the price of labor factors at 1, and the wage rate is given exogenously. The intermediate product production function is set as $x_i = l_i^{\alpha_i} E_i^{1-\alpha_i}$, $\alpha_i \in (0,1)$. The production cost of intermediate products is $l_i + E_i$; as mentioned above, after the VC enters the enterprise, the innovation cost per unit is $\varepsilon$ ($\geq 1$). The ratio of labor to innovation is $l_i/E_i = \varepsilon\alpha_i/(1-\alpha_i)$. According to the principle of cost minimization, the unit cost of producing intermediate products can be derived as $c_i = (\frac{1}{\alpha_i})^{\alpha_i}(\frac{\varepsilon}{1-\alpha_i})^{1-\alpha_i}\left[\frac{1-\alpha_i}{\varepsilon} + \alpha_i\right]$, $\partial c_i/\partial \varepsilon > 0$, that is, $c_i = (\frac{1}{\alpha_i})^{\alpha_i}(\frac{a\phi}{1-\alpha_i})^{1-\alpha_i}\left[\frac{1-\alpha_i}{a\phi} + \alpha_i\right]$, $\partial c_i/\partial \varphi > 0$.

Enterprises need to invest in intermediate products to produce final products. Given the innovation investment of enterprises, the final product production function is $x_i = \alpha\varphi f(q_i)/\gamma_i$, where it is assumed $f(q_i) = q_i^2$. The firm's variable cost of producing $q$ units of final product is $c_i x_i = c_i \alpha\varphi q_i^2/\gamma_i$. In addition, the firm also has to pay fixed costs $\eta f$, $\eta$ are constant.

In order to carry out production and innovation, enterprises need external financing. VC affects the degree of financing constraints of enterprises. When VC enters, the more financing enterprises can obtain. $1 - 1/\varepsilon$ is used to represent the degree of financing constraints; the smaller $\varepsilon$, the smaller the degree of financing constraints. We assume that the external financing needs of enterprises $d_i$ are given. The enterprise must repay the loan at the end of each period, so financing constraints can directly affect the enterprise's innovation investment decision. The profit maximization problem faced by enterprises under financing constraints is as follows:

$$\max\left(\frac{p_i^{1-\sigma}R}{P^{1-\sigma}}\right) - \frac{c_i\alpha\varepsilon q_i^2}{\gamma_i} - \alpha\varepsilon\gamma_i - \eta f \tag{3}$$

$$s.t.(1 - \frac{1}{\varepsilon})\left[\frac{p_i^{1-\sigma}R}{P^{1-\sigma}} - \frac{c_i\alpha\varepsilon q_i^2}{\gamma_i} - (1-d_i)\alpha\varepsilon\gamma_i - \eta f\right] \geq \alpha\varepsilon d_i\gamma_i \tag{4}$$

According to the Kuhn-Tucker condition, when the budget constraint (4) is a tight constraint, the Lagrange multiplier $\lambda > 0$. To ensure $\lambda > 0$, given the initial productivity level of the enterprise and the external financing needs of the enterprise, $\varepsilon$ must be greater than the critical value $\bar{\varepsilon}$. We only discuss the situation where the budget constraint (4) is a tight constraint, that is, the enterprise is constrained by financing. Solving the profit maximization problem, combining the zero profit condition and the market clearing condition, the equilibrium solution (5) can be obtained, where $\Delta = d_i\lambda(\varepsilon - 1)/(1 + \lambda)$, given $d_i \in (0,1)$, $\lambda > 0$, so $\partial\Delta/\partial\varepsilon > 0$, $\varepsilon \geq 1$.

Optimal innovation investment:

$$\gamma_i^* = \left(\frac{\sigma - 1}{2\sigma}\right)^\sigma \frac{R}{P^{1-\sigma}} \frac{1}{\sqrt{1 + \Delta}^{\sigma+1}} \frac{1}{\sqrt{c_i}^{\sigma-1}} \frac{1}{(\alpha\varepsilon)^\sigma} \tag{5}$$

Let us analyse how VC affects TIC by taking the partial derivative of Equation (3) as a constant.

$$\frac{\partial \gamma_i^*}{\partial \varepsilon} = \left(\frac{\sigma-1}{2\sigma}\right)^\sigma \frac{R}{P^{1-\sigma}} \frac{1}{(\alpha\varepsilon)^\sigma} \left[ \underbrace{\underbrace{\frac{1-\sigma}{2} \frac{1}{\sqrt{c_i}^{\sigma+1}} \frac{1}{\sqrt{1+\Delta}^{\sigma+1}} \frac{\partial c_i}{\partial \varepsilon}}_{>0}}_{\text{the internal incentive effect}} - \underbrace{\underbrace{\frac{1+\sigma}{2} \frac{1}{\sqrt{c_i}^{\sigma-1}} \frac{1}{\sqrt{1+\Delta}^{\sigma+3}} \frac{\partial \Delta}{\partial \varepsilon}}_{<0}}_{\text{the external constraint effect}} \right] > 0 \quad (6)$$

$$\frac{\partial \gamma_i^*}{\partial \varepsilon} = \left(\frac{\sigma-1}{2\sigma}\right)^\sigma \frac{R}{P^{1-\sigma}} \frac{1}{(\alpha\varepsilon)^\sigma} \left[ \underbrace{\underbrace{\frac{1-\sigma}{2} \frac{1}{\sqrt{c_i}^{\sigma+1}} \frac{1}{\sqrt{1+\Delta}^{\sigma+1}} \frac{\partial c_i}{\partial \varepsilon}}_{>0}}_{\text{the internal incentive effect}} - \underbrace{\underbrace{\frac{1+\sigma}{2} \frac{1}{\sqrt{c_i}^{\sigma-1}} \frac{1}{\sqrt{1+\Delta}^{\sigma+3}} \frac{\partial \Delta}{\partial \varepsilon}}_{<0}}_{\text{the external constraint effect}} \right] > 0$$

Equation (6) indicates that VC affects TIC in at least two ways. The first effect is the internal incentive effect. VC entry ensures more R&D funds, scientific management, and executive salaries, thereby contributing positively to the TIC (see the first term on the right side of Equation (6)). The second is the external constraint effect: the certification role of VC reduces information asymmetry, communicates active signals to investors, and attracts more external financing. The follow-up easing of constraint effects, primarily that of financing constraint, leads to a higher TIC. (See the second term on the right side of Equation (6)).

Therefore, two hypotheses are proposed in the study.

**Hypothesis 1 (H1).** *VC leads to a higher TIC.*

**Hypothesis 2 (H2).** *VC adds to TIC by producing internal incentive and external constraint easing effects.*

**Hypothesis 2.1 (H2.1).** *VC adds to TIC by spending more on R&D.*

**Hypothesis 2.2 (H2.2).** *VC adds to TIC by giving incentive pay to executives.*

**Hypothesis 2.3 (H2.3).** *VC adds to TIC by diminishing financing constraints.*

**4. Econometric Model, Variables, and Data Specifications**

*4.1. Econometric Model Settings*

To examine the effects of VC participation on the innovation activities of invested companies, we use the DID model to investigate. The reason DID is chosen is not only that there may be differences between enterprises with VC entry and those without VC entry but also that enterprises with VC may differ before and after VC entry. If we simply compare companies with VC entry and those without VC entry, then we cannot rule out whether there are differences between the two types of companies before VC participation rather than VC promoting innovation. If we simply compare companies with VC entry in the difference between before and after VC entry, the change in its innovation trend may be derived from other factors in the external environment that change over time rather than the role of VC. Using the DID method to simultaneously control the two differences can better eliminate interference factors and more reliably strip out the effects of VC on enterprise innovation. Because VCs enter the invested companies at different times, we draw on the previous method to establish the following model [30]:

$$Lnpatent_{it} = \beta_0 + \beta_1 vc + \gamma X_{it} + \lambda_i + \lambda_t + \lambda_j + \varepsilon_{it} \quad (7)$$

where subscripts $i$ and $t$ represent firm and year, respectively; $lnpatent_{it}$ for total firm innovation; $vc$ for the dummy variable of VC entry; $\lambda_i$ and $\lambda_t$ for the fixed effects of the firm and year, respectively; $\lambda_j$ for the fixed effect of the industry; $\varepsilon_{it}$ for the stochastic disturbance term; and $X_{it}$ for a series of controlled variables. See Table 1 for specific definitions.

**Table 1.** Definitions of main variables.

| Variable Type | Variable Name | Variable Definition | Calculation Method |
|---|---|---|---|
| Explained variable | Lnpatent | TIC | The logarithm of the total patent applications |
| | VC | VC | Interaction term: Treat$_i$ × After$_t$ |
| Explanatory variable | Treat | VC entry or none | VC entry (1) or none (0) |
| | After | Pre-and-post-VC entry | Post-entry (1) or not (0) |
| | RDspend | R&D spending | The logarithm of annual R&D spending |
| Mediator variable | RDperson | R&D personnel | The logarithm of annual R&D human capital input |
| | Ave_salary | Average salary of directors, executives, and supervisors | Gross salary divided by the number of directors, executives, and supervisors |
| | SA | Financing constraint | SA = −0.737 size + 0.043 size$^2$ − 0.04 age |
| | Age | Firm age | The current year − The opening year + 1 |
| | Ownership | Firm ownership | State-owned or state-controlled firm (1), or not (0) |
| Controlled variable | Size | Firm size | The logarithm of the gross assets of the current year |
| | ROE | Return on assets | Gross assets divided by the net profit ratio |
| | Debt | Debt-to-assets ratio | Gross assets divided by gross debts |
| | PPE | proportion of fixed assets | Net fixed assets divided by gross fixed assets |
| | Intensity | Capital intensity | Fixed assets divided by total employees |

### 4.2. Variable Selection and Calculation Description

#### 4.2.1. Explained Variable: TIC (Lnpatent$_{it}$)

As with Song et al. [4], we measure TIC based on the number of patent applications for three reasons. First, the number of applications belongs to the innovation output and can better reflect the firm's TIC than investments in R&D and other innovation spending indicators. Second, because the number of applications has a shorter time lag than that of patents granted, the number of applications can better reflect the firm's current innovation and competitiveness [31]. Third, under the Chinese Patent Act, patents fall under three categories: invention, utility model, and industrial design, all of which belong to IP rights. Therefore, the total number of the three types of patents used as a dependent variable will be more representative [32]. Moreover, some firms have no applications; therefore, we take the logarithm of 1 plus the total number of applications to be used as the explained variable.

#### 4.2.2. Core Explanatory Variable: VC

In this study, the interaction term Treat$_i$ ×After$_t$ was constructed to represent VC. The core explanatory variable, Treat$_i$, refers to VC entry. If there is VC entry, we set its value as 1 and define it as a treatment group of 1.501 firms; if there is no VC entry, we set its value as 0 and define it as a control group of 2492 firms. After$_t$ is the pre- and post-entry virtual variable. The value is 1 if VC enters the current year or later, and 0 if otherwise. To define a VC firm, we refer to the treatment methods of Wu et al. and Song et al. [4,33]. First, we gather information of approximately 10 largest shareholders in the prospectus of listed companies. If the name of any shareholder contains 'VC', 'VC investment', or 'VC investing', the company is considered a VC firm. Second, we check the shareholders' information disclosures. If a shareholder's principal business is VC, the company is considered a VC firm. Third, we evaluate the firms in PEDATA based on the keyword 'institutional investor'. Finally, we use Stata 16.0 for the VC data processing.

### 4.2.3. Controlled Variables

We refer to the findings of Chen et al. and Lu et al. [30,34] by selecting firm age (Age), ownership (Ownership), size (Size), return on assets (ROA), debt-to-assets ratio (Debt), proportion of fixed assets (PPE), and capital intensity (Intensity) as controlled variables to better measure the effects of VC (VC) on TIC (Lnpatent). See Tables 1 and 2 for definitions of specific variables and relevant descriptive statistics.

**Table 2.** Descriptive statistics of the main variables.

| Variables | Obs | Mean | SD | Min | Max |
|---|---|---|---|---|---|
| Lnpatent | 28,892 | 3.232 | 2.145 | 0.00 | 11.03 |
| VC | 28,892 | 0.303 | 0.459 | 0.00 | 1.00 |
| Age | 28,892 | 16.882 | 6.185 | 0.00 | 62.00 |
| Ownership | 28,892 | 0.373 | 0.484 | 0.00 | 1.00 |
| Size | 28,892 | 22.117 | 1.357 | 0.00 | 28.64 |
| ROA | 28,892 | 0.041 | 0.060 | −0.24 | 0.20 |
| Debt | 28,892 | 0.415 | 0.207 | 0.05 | 0.88 |
| PPE | 28,892 | 0.212 | 0.162 | 0.00 | 0.95 |
| Intensity | 28,892 | 2.448 | 1.933 | 0.40 | 12.34 |

### 4.3. Data Description

Data are from companies that went public on the A-stock market from 2010 to 2020. The reasons for selecting A-share listed companies as the sample are as follows. Firstly, the research topic of this paper is on Chinese enterprises. Listed companies in China are relatively standardized operation of enterprises, which is the standard of general enterprise operation. In addition, compared with other Chinese stock markets (e.g., B-share or H-share), in the A-share market, the number of listed companies is the largest, so a sufficient sample size can be guaranteed. So, A-share data is selected as the sample is representative. Secondly, enterprises need to meet different conditions for listing in different stock markets. The selection of A-share sample makes more similarities between different enterprises and alleviates the impact of enterprise heterogeneity on the empirical results. Finally, due to the availability of experimental data, the information disclosure of A-share listed companies is more complete, which is more conducive to the experiment. In addition, VC data are author-selected, while patent and financial data of listed companies are obtained from the CSMAR database. Prior to analysis, four types of data were deleted. The first concerns listed companies that do not disclose R&D spending. The second concerns listed financial companies. The third concerns ST- and * ST-listed companies. The fourth concerns listed companies with data deficiency. The resulting unbalanced panel data include 28,892 observed values for 3993 listed companies.

### 4.4. Descriptive Statistics

Table 2 provides the descriptive statistics of variables. The mean value of VC was 0.303, indicating that only a portion of the investees was entered by VC. The explained variable, *lnpatent*, has a maximum value of 11.03 and mean value of 3.232, indicating a major difference in *lnpatent* between listed companies.

## 5. Empirical Analysis and Results

### 5.1. Baseline Regression

Table 3 provides the baseline regression results for the effects of VC on TIC. Column 1 presents the regression results with no controlled variable and shows that the regression coefficient is 0.893, which is significantly positive at the 1% level. This testifies that VC significantly improves TIC. We further control for the effects of factors such as age, ownership, and size and operating status factors such as debt and intensity on *lnpatent*. In Column 2, the regression results are combined with the correlated controlled variables; in Columns 3 and 4, the regression results are combined with time- and industry-fixed effects. As the

regression results show, VC remains significantly positive at the 1% level. This supports H1. There are two reasons. First, the TIC is a significant symbol of enterprise value. VC can improve TIC to derive excess earnings from the investee. Second, resources, experience, information, and positive investment signals lead to VC.

**Table 3.** Baseline regression results.

| Variables | (1) Lnpatent | (2) Lnpatent | (3) Lnpatent | (4) Lnpatent |
|---|---|---|---|---|
| VC | 0.8930 *** | 0.2452 *** | 0.1752 *** | 0.1660 *** |
| | (0.026) | (0.029) | (0.028) | (0.027) |
| Age | | 9.0409 *** | 1.0024 *** | 0.9201 *** |
| | | (0.390) | (0.256) | (0.255) |
| Ownership | | −0.0024 | 0.1647 * | 0.1687 * |
| | | (0.099) | (0.096) | (0.096) |
| Size | | 0.1426 *** | 0.0525 ** | 0.0692 *** |
| | | (0.027) | (0.021) | (0.021) |
| ROA | | −0.2966 *** | −0.2616 ** | −0.2400 ** |
| | | (0.113) | (0.107) | (0.106) |
| Debt | | 0.3561 *** | 0.2991 *** | 0.2994 *** |
| | | (0.071) | (0.066) | (0.065) |
| PPE | | 0.4710 *** | 0.3296 *** | 0.3170 *** |
| | | (0.090) | (0.086) | (0.084) |
| Intensity | | 0.0298 | 0.0026 | 0.0055 |
| | | (0.026) | (0.037) | (0.037) |
| Constant | 2.9621 *** | −1.7581 *** | 0.7704 * | 0.3661 |
| | (0.008) | (0.559) | (0.432) | (0.607) |
| Individual FE | YES | YES | YES | YES |
| Year FE | NO | NO | YES | YES |
| Industry FE | NO | NO | NO | YES |
| Observations | 28,892 | 28,892 | 28,892 | 28,892 |
| Number of id | 3993 | 3993 | 3993 | 3993 |
| Within_R2 | 0.1276 | 0.3888 | 0.4467 | 0.4544 |

Notes: ***, **, and * denote statistical significance at the 1%, 5%, and 10% level, respectively. Standard errors are in parentheses.

*5.2. Parallel Trend Hypothesis Test*

Prior to using the DID method to analyse the effects of VC on TIC, it is necessary to ensure that the model complies with the parallel trend; for example, there must be no significant difference between treatment and control groups prior to the implementation of the policy. Therefore, we need to conduct a parallel trend test to determine whether treatment and control groups prior to VC have similar characteristics. In this study, a parallel trend test was conducted based on the event study by Jacobson et al. [35]. The model is as follows:

$$Lnpatent_{it} = \beta_0 + \sum_{t=-8}^{t=8} \beta_t R_t + \gamma X_{it} + \lambda_i + \lambda_t + \lambda_j + \varepsilon_{it} \tag{8}$$

where $R_t$ represents the $t$(th) year of entry. In this study, we used the year preceding entry as the base period to determine the parallel trend test results, as shown in Figure 1. The dotted lines represent the estimates of $\beta_t$ at the 95% confidence interval and coefficient. Therefore, prior to entry, the coefficient wavers around zero. No significant difference existed before entry, and there was compliance with the parallel trend. In the second year of entry, VC had a significantly positive effect on TIC. As time passed, the estimated coefficient increased in value annually and became insignificant in the fifth year of entry. These characteristics indicate that VC did not contribute immediately to TIC upon entering the firm, a phenomenon in line with the uncertainty and time lag effect inherent in TIC activities. Moreover, while VC can contribute positively and continuously to TIC upon entry for a

time, its time effects on TIC begin to emerge in the course of time. The possible reason is that the longer the cooperation between the VC firm and investee, the greater the likelihood of a conflict of interest; for example, the VC firm wants the investee to further enhance TIC, and the investee management desires to invest more resources in market performance.

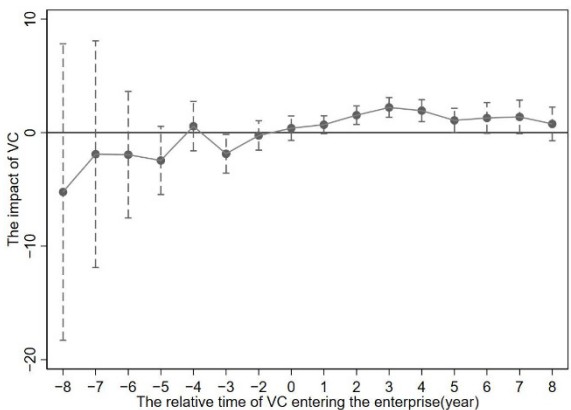

**Figure 1.** Parallel trend hypothesis.

### 5.3. Influence Mechanism Test

As demonstrated in the foregoing baseline regression results, VC can contribute significantly to TIC. To further investigate the mechanism through which VC acts on TIC, we used the foregoing theoretical mechanism and hypotheses to conclude that VC impacts TIC by enhancing the internal incentive and easing the external constraint effects. To test these two effects, this study draws on Song and Zheng [36] to test the mechanism of VC on enterprises' technological innovation.

First, we tested how the mediator variables affect investment in R&D by building the following econometric model:

$$M_{it} = \beta_0 + \beta_1 VC_{it} + \gamma X_{it} + \lambda_i + \lambda_t + \lambda_j + \varepsilon_{it} \tag{9}$$

Second, we tested how VC and the mediator variables affect TIC by building the following econometric model:

$$Lnpatent_{it} = \beta_0 + \beta_1 vc_{it} + \beta_2 M_{it} + \gamma X_{it} + \lambda_i + \lambda_t + \lambda_j + \varepsilon_{it} \tag{10}$$

In Equations (9) and (10), $M_{it}$ stands for mediator variables, such as investment in R&D, executive incentives, and financing constraints.

#### 5.3.1. Investment in R&D

VC brings sufficient funds and professional teams, thereby further adding to investment in R&D; meanwhile, capital intensity has a direct effect on TIC. Specifically, VC has a capital increase effect that brings more liquid capital and, hence, more R&D funds. In addition, VC has a supervision and governance effect because VC firms with extensive management experience supervise scientific disbursement through the invention of R&D resources to better safeguard return on investment (ROI) [37]. This explains why VC secures competitive edges of management and resource utilisation for the investee to improve investment in R&D, which is a factor that directly impacts TIC. In this study, the total number of persons involved in R&D activities in a year is used as R&D spending (RDspend), while the number of persons involved per invention patent is used as the human capital input (RDperson) [32]. In Table 4, Columns 1 (RDPerson) and 3 (RDspend) are used as explained variables to conduct the empirical study, and it is found that VC has a significantly positive effect on investment in R&D at the 1% level. This means that VC has a significantly positive effect on R&D spending and human capital inputs. As Columns 2 and 4 show, VC has a significantly positive effect on TIC. However, the inclusion of mediator variables, such as

investment in R&D, weakens the effect of VC on TIC to some degree, which supports H2.1 that investment in R&D is important for VC to affect TIC.

**Table 4.** Mechanism test for investment in R&D.

| Variables | (1) RDPerson | (2) Patent | (3) RDspend | (4) Lnpatent |
|---|---|---|---|---|
| VC | 0.1207 *** | 0.0742 *** | 0.6173 *** | 0.1628 *** |
| | (0.003) | (0.013) | (0.105) | (0.013) |
| RDPerson | | 0.8365 *** | | |
| | | (0.024) | | |
| RDspend | | | | 0.0051 *** |
| | | | | (0.001) |
| Age | −0.2553 *** | 1.2160 *** | −8.0052 *** | 0.9607 *** |
| | (0.058) | (0.222) | (1.761) | (0.226) |
| Ownership | 0.0225 * | 0.1459 *** | −0.7498 * | 0.1725 *** |
| | (0.013) | (0.049) | (0.394) | (0.050) |
| Size | 0.0479 *** | 0.0124 | 1.0136 *** | 0.0641 *** |
| | (0.002) | (0.008) | (0.064) | (0.008) |
| ROA | 0.0321 * | −0.2885 *** | 0.3016 | −0.2415 *** |
| | (0.019) | (0.071) | (0.563) | (0.072) |
| Debt | 0.0628 *** | 0.2466 *** | −3.1421 *** | 0.3153 *** |
| | (0.009) | (0.034) | (0.273) | (0.035) |
| PPE | 0.0438 *** | 0.2930 *** | 0.1558 | 0.3162 *** |
| | (0.012) | (0.044) | (0.356) | (0.046) |
| Intensity | 0.0026 | 0.0005 | −0.2030 | 0.0065 |
| | (0.007) | (0.026) | (0.206) | (0.026) |
| Constant | −1.0056 *** | 1.6116 *** | −13.6015 *** | 0.4350 * |
| | (0.043) | (0.168) | (1.936) | (0.248) |
| Individual FE | YES | YES | YES | YES |
| Year FE | YES | YES | YES | YES |
| Industry FE | YES | YES | YES | YES |
| Observations | 28,892 | 28,892 | 28,892 | 28,892 |
| Number of id | 3993 | 3993 | 3993 | 3993 |
| Within_R2 | 0.7535 | 0.4721 | 0.2479 | 0.4552 |

Notes: *** and * denote statistical significance at the 1% and 10% level, respectively. Standard errors are in parentheses.

### 5.3.2. Executive Incentives

Modern firms generally have shareholders who are independent of management. Management often faces moral hazards regarding technological innovation-related decision-making; therefore, technological innovation is reduced to avoid these hazards. However, this will indirectly detract from shareholders' long-term interests. As a professional financial intermediary intended to materialise high returns, the VC firm enters the investee, optimises the corporate governance structure, and implements executive incentive pay programs that effectively inhibit shareholder-management friction and management's risk aversion while intensifying management's motive to engage in investment programs, particularly when it concerns the R&D division [38]. By supervising the executive, VC lowers the agency cost of technological innovation-related decision-making while heightening TIC. Therefore, we introduce the average salary of directors, senior executives, and supervisors as incentive indicators to conduct a study based on the research method of Huang and Wang [39]. As shown in the regression results in Column 1 of Table 5, when the average salary of the current year is used as an explained variable, the regression coefficient of VC is significantly positive at the 5% level. As the empirical findings in Column 2 show, the regression coefficient of VC is significantly positive at the 1% level and lower than the results of the baseline regression. Executive incentives have a partial mediation effect, which supports H2.2.

**Table 5.** Executive incentive and financing constraint mechanism test.

| Variables | (1) Av_salary | (2) Lnpatent | (3) SA | (4) Lnpatent |
|---|---|---|---|---|
| VC | 0.0166 ** | 0.1651 *** | −0.5376 *** | 0.1647 *** |
| | (0.008) | (0.013) | (0.194) | (0.028) |
| Av_salary | | 0.0507 *** | | |
| | | (0.020) | | |
| SA | | | | −0.0023 * |
| | | | | (0.001) |
| Age | −0.3854 *** | 0.9396 *** | −514.9573 *** | −0.2672 |
| | (0.069) | (0.226) | (21.597) | (0.617) |
| Ownership | 0.0022 | 0.1686 *** | −5.2404 *** | 0.1566 |
| | (0.034) | (0.050) | (1.394) | (0.097) |
| Size | 0.0906 *** | 0.0646 *** | −0.3930 | 0.0683 *** |
| | (0.011) | (0.008) | (0.426) | (0.021) |
| ROA | 0.3582 *** | −0.2581 *** | 6.8886 *** | −0.2241 ** |
| | (0.035) | (0.073) | (1.083) | (0.106) |
| Debt | −0.0120 | 0.3000 *** | 2.7688 *** | 0.3058 *** |
| | (0.020) | (0.035) | (0.717) | (0.065) |
| PPE | −0.0206 | 0.3180 *** | 1.1574 ** | 0.3197 *** |
| | (0.022) | (0.046) | (0.538) | (0.084) |
| Intensity | −0.0151 ** | 0.0062 | −0.1114 | 0.0052 |
| | (0.007) | (0.026) | (0.098) | (0.037) |
| Constant | −1.7243 *** | 0.4535 * | 70.9839 *** | 0.5297 |
| | (0.245) | (0.250) | (8.831) | (0.615) |
| Individual FE | YES | YES | YES | YES |
| Year FE | YES | YES | YES | YES |
| Industry FE | YES | YES | YES | YES |
| Observations | 28,892 | 28,892 | 28,892 | 28,892 |
| Number of id | 3993 | 3993 | 3993 | 3993 |
| Within_R2 | 0.2655 | 0.4545 | 0.8416 | 0.4545 |

Notes: ***, **, and * denote statistical significance at the 1%, 5%, and 10% level, respectively. Standard errors are in parentheses.

### 5.3.3. Financing Constraint

As a third party, the VC firm is considered by external financiers as a signal of favourable firm operations; therefore, other external investors often tend to follow the VC firm in participating in financing activities. This explains why VC can improve the information asymmetry typical of financing activities, reduce corporate financing costs, and obtain more R&D funds, thereby successfully driving TIC up. In this study, the SA index is used as a proxy variable for financing constraints. The greater the SA index, the more serious the financing constraint a firm faces (see Equation (11) for details) [40]. As indicated in Column 3 of Table 5, VC has a significantly negative effect on financing constraints at the 1% level; for example, VC can relieve financing constraints. As shown in the regression results in Column 4, financing constraints have a significantly negative effect on TIC at the 10% level; for example, financing constraints inhibit innovation activities. The regression coefficient of VC is smaller than the results of the baseline regression, indicating that financing constraints weaken the effect of VC on TIC, as proposed in H2.3. In other words, financing constraints are an important way in which VC affects TIC.

$$SA = -0.737Size + 0.043Size^2 - 0.04Age \tag{11}$$

### 5.4. Heterogeneity Test

As the next step, we analyse the effects of VC on TIC from the perspectives of heterogeneity as firm ownership (state-owned firm or VC_state and non-state-owned firm or VC_others), size (firms above the designated size or VC_big and others or VC_small), and

technological sophistication (high-tech firm or VC_hightech and medium- and low-tech firms or VC_lowtech).

### 5.4.1. Firm Ownership

In this study, listed companies are divided into state-owned and non-state-owned firms to study the differential effect of VC on TIC as a result of differential TIC. Column 1 of Table 6 shows the regression results. The empirical findings prove that the regression coefficient of ownership is significantly positive at the 1% level. The regression coefficient of state-owned firms (VC_state) is greater than that of non-state-owned firms (VC_others), indicating that VC has a more significant effect on TIC enhancement. There are two possible reasons for this result. First, state-owned firms are typically larger than non-state-owned firms and attract VC joint investment more easily; second, state-owned firms have an inherent political bond with the government that tends to secure government policy privileges (e.g., fiscal subsidy, tax preference, and bank credit) which leads to VC success in technological innovation [41].

**Table 6.** Heterogeneity test.

| Variables | (1) Ownership | (2) Scale | (3) Sophistication | (3) Area |
|---|---|---|---|---|
| VC_state | 0.2099 *** (0.019) | | | |
| VC_others | 0.1504 *** (0.017) | | | |
| VC_big | | 0.2224 *** (0.015) | | |
| VC_small | | 0.0936 *** (0.017) | | |
| VC_hightech | | | 0.2088 *** (0.019) | |
| VC_lowtech | | | 0.1515 *** (0.016) | |
| VC_east | | | | 0.2470 *** (0.033) |
| VC_west | | | | 0.1480 *** (0.016) |
| VC_med | | | | 0.2275 *** (0.029) |
| Age | 1.0649 *** (0.228) | 1.1776 *** (0.228) | 0.9882 *** (0.227) | 0.9194 *** (0.352) |
| Ownership | 0.1487 *** (0.051) | 0.1566 *** (0.051) | 0.1636 *** (0.051) | 0.0125 (0.020) |
| Size | 0.0534 *** (0.008) | 0.0412 *** (0.008) | 0.0523 *** (0.008) | 0.0516 *** (0.008) |
| ROA | −0.2660 *** (0.073) | −0.2694 *** (0.072) | −0.2592 *** (0.073) | −0.2625 *** (0.073) |
| Debt | 0.3059 *** (0.035) | 0.2927 *** (0.035) | 0.2957 *** (0.035) | 0.3019 *** (0.035) |
| PPE | 0.3322 *** (0.045) | 0.3317 *** (0.045) | 0.3291 *** (0.045) | 0.3299 *** (0.045) |
| Intensity | 0.0027 (0.027) | 0.0023 (0.027) | 0.0027 (0.027) | 0.0030 (0.027) |
| Constant | 0.7516 *** (0.170) | 1.0096 *** (0.173) | 0.7769 *** (0.170) | 0.8426 *** (0.169) |
| Individual FE | YES | YES | YES | YES |
| Year FE | YES | YES | YES | YES |
| Industry FE | YES | YES | YES | YES |
| Observations | 28,892 | 28,892 | 28,892 | 28,892 |
| Number of id | 3993 | 3993 | 3993 | 3993 |
| Within_R2 | 0.4468 | 0.4480 | 0.4469 | 0.4467 |

Notes: *** denote statistical significance at the 1% level. Standard errors are in parentheses.

### 5.4.2. Firm Size

Based on the median of the firms' current year total assets, they are classified into firms above the designated size (VC_big) and other firms (VC_small). As shown in the regression results in Column 2 of Table 6, both types of firms prove significantly positive, which confirms the robustness of the analysis results. Furthermore, VC is more conducive to TIC than other firms. There are two reasons. First, firms above a certain size have the advantage of better internal governance mechanisms; therefore, compared with other types of firms, more appropriate resource allocation and management systems contribute positively to the role of VC in enhancing TIC. Second, as other types of firms are comparatively deficient in funds and resources, they have to bear a heavier innovation cost burden; notwithstanding VC entry, they will fail to improve their TIC within a short period.

### 5.4.3. Technological Sophistication

To investigate whether the effects of VC on TIC vary with technological sophistication, we introduce the treatment method of Lv et al. [42] by classifying industries into high-tech (VC_hightech) and medium- and low-tech industries (VC_lowtech) at the technological level. In this study, eight industries (professional technical service, special equipment manufacturing, the Internet and related service, instrument and meter manufacturing, chemical raw materials and chemical products manufacturing, electrical machinery and equipment manufacturing, computer communications and other electronic equipment manufacturing, software, and information technology services) are defined as high-tech industries. Other industries are defined as medium- and low-tech industries. There are two possible reasons for this finding. First, high-tech firms feature high technology as their core competitive advantage; therefore, they have to introduce continual TIC updates to grab more market shares. Second, the industries which high-tech firms belong to normally attract more innovation talent and, hence, a bigger innovation environment. This explains why VC has an advantage in TIC enhancement for investees.

### 5.4.4. Area Distribution

To further analyze whether the influence of VC on enterprise TIC varies, the provinces are divided into the eastern, central, and western regions, including Beijing, Fujian, Guangdong, including Guangdong, Hainan, Jiangsu, Liaoning, Shanghai, Tianjin and Shanxi provinces, and the rest in western regions. The regression results are shown in Table 6, column (4), and all three regions have promoted the improvement of the TIC of the enterprises, but the enterprises in the eastern regions are the most significant. We believe that the possible reason is that, compared with the central and western regions, the eastern region economic development level is higher, can gather more scientific and technological innovation resources and high-end talents, and can attract more high-quality risk investment institutions. On the whole this is conducive to promote district enterprises to further improve the level of technological innovation and coupled with the eastern region's relatively perfect legal environment can ensure the VC institutions to the greatest extent. Therefore, the TIC of enterprises in the eastern region is more significant.

### *5.5. Robustness Test*
### 5.5.1. Placebo Test

A placebo test is conducted to test and estimate whether the policy effect is ascribable to other unobservable factors. Since VC firms in the treatment group entered at different times, the placebo test imagines a treatment group and VC entry time. First, we set aside one-year data for all listed companies. We sampled 300 firms as a pseudo-group and the rest as the control group to ensure that the independent variables (VC) resulting from random sampling have no effect on TIC. Second, we sampled a random year as the VC entry time. Third, the two steps are repeated 500 times before conducting the regression, as per Equation (7). Any significant discovery in the placebo test indicated deviation from the regression results in this study. The results of the placebo test are shown in Figure 2. The red

curve represents the kernel density of the coefficients, the blue dots make up the coefficient scatterplot, and the red line perpendicular to the x-axis represents the coefficients of the baseline regression. Figure 2a (OLS estimates) and Figure 2b (industry- and time-fixed effects) have the same placebo test results; that is, the average value of the coefficients of post-repetition VC is almost 0, which significantly differs from the estimated coefficients in Column 2 of Table 3, which result from baseline regression. In addition, most of the p-values were greater than 0.1, indicating that the random test results were not significant. Therefore, the effects of VC on TIC are not driven by any unobservable driving factors.

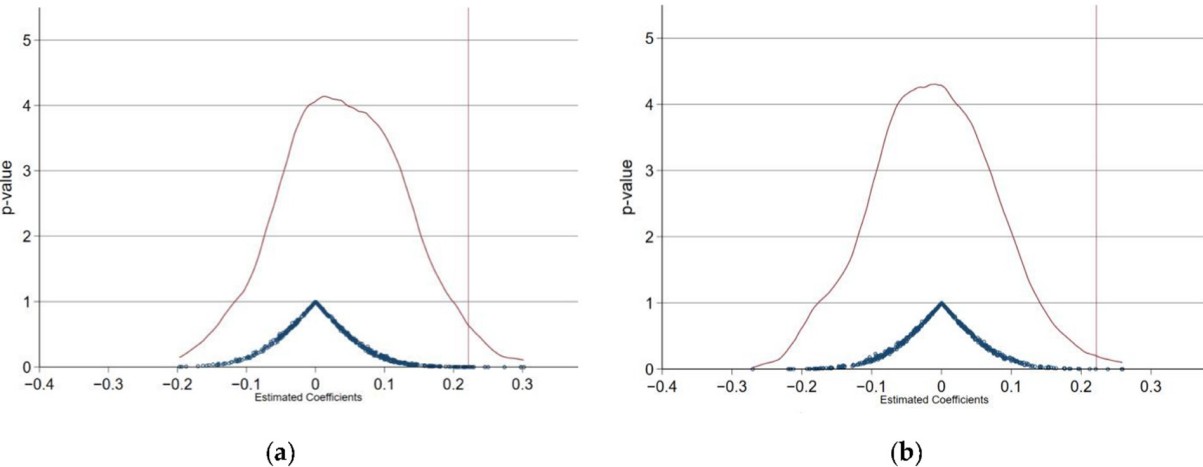

**Figure 2.** Placebo test. (**a**) OLS estimates. (**b**) FE estimates.

### 5.5.2. Propensity Score Matching (PSM) Analysis

The PSM method is used in this study to match firms with characteristics similar to those of the treatment group in more than one dimension to determine the net effect of VC on TIC [43]. The matching variables include the net profit ratio (yyjll), debt-to-equity ratio (cqbl), net sales-to-assets ratio (zzczzl), fixed assets ratio (PPE), firm age (Age), and number of executives (ExecutivesNumber). The pre-matching results are shown in Figure 3a. The lateral axis represents the propensity score, the vertical axis represents kernel density, the solid line represents the treatment group, and the dotted line represents the control group. Overall, the treatment group had a higher propensity score than the control group, and they were similarly distributed. Figure 3b illustrates the nearest neighbour matching (NNM) results and compares them with Figure 3a, where the deviation between the control and treatment groups is corrected with a comparatively ideal effect.

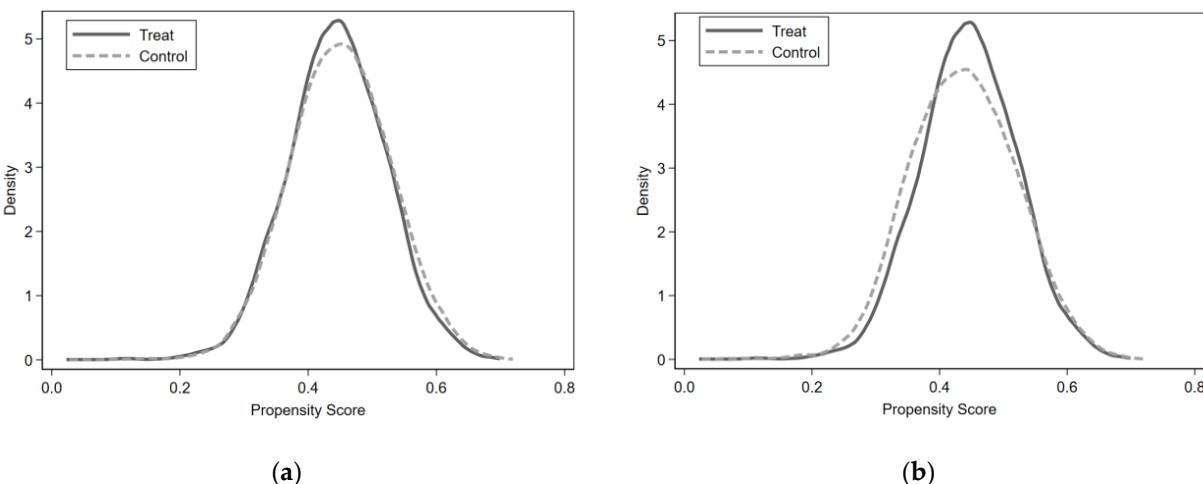

**Figure 3.** PSM analysis. (**a**) Pre-matching probability distribution graph. (**b**) Post-matching probability distribution graph.

Table 7 reports the NNM-based regression results, which show that the VC coefficient is significantly positive at the 1% level and is identical to the baseline regression results in Column 2 of Table 3, demonstrating the robustness of the research conclusions.

**Table 7.** The NNM-based regression results.

| Variables | (1) | (2) |
|---|---|---|
| | **Lnpatent** | **Lnpatent** |
| VC | 0.8930 *** | 0.1779 *** |
| | (0.026) | (0.028) |
| yyjll | | 0.0020 *** |
| | | (0.001) |
| cqbl | | 0.0012 |
| | | (0.001) |
| zzczzl | | −0.0493 * |
| | | (0.026) |
| PPE | | 0.4618 *** |
| | | (0.083) |
| ExecutivesNumber | | 0.0068 * |
| | | (0.004) |
| Age | | 13.7765 *** |
| | | (0.312) |
| Constant | 2.9621 *** | 0.6337 *** |
| | (0.008) | (0.091) |
| Observations | 28,892 | 28,434 |
| R-squared | 0.1276 | 0.4270 |
| Number of id | 3993 | 3878 |
| Within_R2 | 0.1276 | 0.4270 |
| Between_R2 | 0.0731 | 0.0168 |
| Overall_R2 | 0.1032 | 0.0012 |

Notes: ***, and * denote statistical significance at the 1%, and 10% level, respectively. Standard errors are in parentheses.

### 5.5.3. Variable Substitution

In the foregoing context, the number of patent applications was used as an explained variable. We then selected two substitution variables, the number of invention patents and R&D intensity, to test the robustness of the regression results. We refer to the research method of Xia and Le [44] and take the logarithm of the number of the firm's invention patents in the current year (Lninvent) as the dependent variable, as an invention patent has a higher technological level and therefore has a greater impact on TIC. Column 1 of Table 8 presents the empirical findings; the VC coefficient is significantly positive at the 1% level. Song et al.'s [4] research method was used to measure TIC based on the ratio of R&D spending to operating income (RDSpendSumRatio). As Column 2 of Table 8 shows, the coefficient of the explanatory variable is significantly positive at the 5% level. Therefore, as with the conclusions of previous studies, VC can substantially improve TIC.

### 5.5.4. Instrumental Variable Method

We consider that VC may be attracted by companies that have patents before VC enters, so there may be endogenous problems. To ensure the reliability of the regression results, the instrumental variable method was introduced for an endogeneity test. As proved by Lu et al. [34], VC has an evident trend of 'local preference'; for example, regional VC intensity directly impacts investment probability. As stated by Zhang and Zhang [45], this is because information asymmetry and agency problems increase with distance, causing the majority of VC firms to invest in neighbouring firms. Referring to Lu et al. [34] and Chen and Zhao [46], we choose two instrumental variables: (1) take the logarithm (LnVC1) after adding one to the number of VC institutions in the province in the year being invested. (2) Density of VC institutions (LnVC2) in the province where the listed company is located,

that is, the number of VC institutions in the province where the company is located is divided by the number of listed companies in the province, and then a logarithm is added. Whether a company is invested is highly correlated with the number of VC institutions in the region, and the number of VC institutions in each province are the only variables based on regions that do not affect the technological innovation ability of enterprises, so they meet the requirements of instrumental variable correlation and exogeneity. The data excludes overseas VC institutions. The regression results are shown in Table 9. The coefficients of the explanatory variables are all significantly positive at the 1% level, which is consistent with the benchmark regression above.

**Table 8.** Variable substitution.

| Variables | (1) | (2) |
|---|---|---|
| | **Lninvent** | **RDSpendSumRatio** |
| VC | 0.1470 *** | 0.4122 ** |
| | (0.014) | (0.160) |
| Age | 1.0240 *** | 9.6894 *** |
| | (0.233) | (2.694) |
| Ownership | 0.2328 *** | −0.0434 |
| | (0.052) | (0.603) |
| Size | 0.0728 *** | 0.5239 *** |
| | (0.008) | (0.098) |
| ROE | −0.0391 | −8.2461 *** |
| | (0.075) | (0.862) |
| Debt | 0.3608 *** | −4.3447 *** |
| | (0.036) | (0.418) |
| PPE | 0.2110 *** | 0.1843 |
| | (0.047) | (0.544) |
| Intensity | −0.0016 | −3.6765 *** |
| | (0.027) | (0.315) |
| Constant | −0.2475 | −8.0667 *** |
| | (0.257) | (2.963) |
| Individual FE | YES | YES |
| Year FE | YES | YES |
| Industry FE | YES | YES |
| Observations | 28,892 | 28,892 |
| Number of id | 3993 | 3993 |
| Within_R2 | 0.3581 | 0.0290 |

Notes: *** and ** denote statistical significance at the 1% and 5% level, respectively. Standard errors are in parentheses.

**Table 9.** Instrument Variables.

| Variables | (1) | (2) |
|---|---|---|
| | **LnVC1** | **LnVC2** |
| VC | 0.0468 *** | 0.1297 *** |
| | (0.003) | (0.024) |
| Age | −0.0195 *** | 0.0348 *** |
| | (0.002) | (0.013) |
| Ownership | 0.0005 | −0.0084 *** |
| | (0.000) | (0.002) |
| Size | 0.0061 *** | −0.0194 ** |
| | (0.002) | (0.010) |
| ROE | 0.0363 *** | 0.0828 *** |
| | (0.003) | (0.014) |

**Table 9.** *Cont.*

| Variables | (1) | (2) |
|---|---|---|
| | LnVC1 | LnVC2 |
| Debt | −0.0004 | 0.0151 *** |
| | (0.001) | (0.004) |
| PPE | 0.0042 *** | 0.0025 |
| | (0.001) | (0.002) |
| Intensity | −0.0000 | 0.0016 *** |
| | (0.000) | (0.000) |
| Constant | 0.0050 | 0.0453 ** |
| | (0.003) | (0.020) |
| Individual FE | No | Yes |
| Year FE | No | Yes |
| Industry FE | No | Yes |
| Kleibergen-Paap rk LM statistic | 357.273 | 28.082 |
| Kleibergen-Paap rk Wald F statistic | 184.728 | 14.055 |
| Hansen J statistic | 255.155 | 6.232 |
| Observations | 28,888 | 28,888 |
| Centered_$R^2$ | −0.3464 | −5.4963 |
| Uncentered_$R^2$ | 0.5885 | −0.9853 |

Notes: *** and ** denote statistical significance at the 1%and 5% level, respectively. Standard errors are in parentheses.

## 6. Conclusions and Limitations

We analysed VC using data on 3993 companies that went public from 2010 to 2020 based on a DID model to systematically analyse the effects of VC on TIC and the relevant mechanism. Studies show that VC can significantly improve TIC. The heterogeneity analysis shows that VC investments in large, state-owned, and high-tech firms work better to improve TIC. VC influences TIC through the medium of not just internal incentives (e.g., more R&D spending and higher incentive pay for executives) but also external constraint easing incentives (e.g., financing constraint easing). A range of robustness tests, including the parallel trend hypothesis, placebo, and instrument variable test, was conducted to verify the validity of the conclusions.

The study's conclusions serve not only to account for the mechanism of how VC works on TIC from more theoretical perspectives but also provide significant reference to empirical VC and TIC activities. First, a firm must introduce external assistance, reallocate resources appropriately, and improve internal management. It is concluded that because VC can influence TIC significantly, a growth company should leverage external resources (e.g., VC) to provide funds and experience for development while enhancing TIC through technology and management. The mechanism analysis shows that VC serves to improve TIC through R&D spending, incentive pay for executives, and financing constraints, among others. In other words, a firm, which has received VC investments, should not be contented with stock trading and financial report figures but should invest more in R&D and human capital. Second, VC firms should improve their business and make every effort to partner with high-performance investees. VC firms should strengthen their post-investment management and participate in corporate supervision and governance to enhance TIC and partake in innovation dividends. Moreover, VC firms should build up industry experience, resources, and reputation by investing intensively in strategic industries and cooperating with state-owned, large, and high-tech firms for higher returns. Finally, policymakers should update VC investing-relevant laws, regulations, and incentive mechanisms so that VC firms and investees hold more closely together. For example, SME-oriented subsidies should be provided, TIC indicators should be updated, and TIC activities should be promoted. In addition, policy makers should also build a good market innovation environment to help enterprises innovate and develop sustainably and enhance the motivation and vitality of enterprise development.



This study has several limitations. First is the limitation of the depth of the study. At the VC level, this study mainly considers the influence of VC on enterprises' TIC. In a follow-up study, the characteristics of VC can be further divided (e.g., the reputation of VC and degree of participation of VC), explored regarding its influence on the TIC of enterprises and the mechanism of action, and then studied more deeply and comprehensively. In addition, owing to the limitations of the data, this study does not consider the impact of the background of VC on the technological innovation of enterprises. The second limitation is the sample size. The subjects in this study are A-share listed companies in China. However, in reality, there are a large number of enterprise samples supported by VC, but they are not listed in the end. For unlisted companies, whether VC is involved and how its involvement affects the technological innovation of enterprises' capacity, this study does not conduct related research. This is because the coverage of the financial data of non-listed companies in domestic and foreign databases is quite limited, and there is a problem of a large amount of missing data, which is not enough to form a good support for this study. Therefore, the applicability of the conclusions of this study is limited.

**Author Contributions:** Conceptualization, Y.S.; methodology, S.J.; software, S.J.; validation, Y.S. and S.J.; formal analysis, Z.L.; investigation, Y.S.; data curation, Y.S.; writing—original draft preparation, S.J.; writing—review and editing, Y.S.; visualization, S.J.; supervision, Z.L.; project administration, Z.L.; funding acquisition, Z.L. All authors have read and agreed to the published version of the manuscript.

**Funding:** This research was funded by financial support of the National Social Science Fund of China (Research Department of Innovation Path of Education for National Unity and Progress in Southwest Frontier in the Internet Age; Project NO: 20BMZ013).

**Institutional Review Board Statement:** Not applicable.

**Informed Consent Statement:** Not applicable.

**Data Availability Statement:** Not applicable.

**Acknowledgments:** The authors thank the editor and anonymous reviewers for their useful comments and suggestions.

**Conflicts of Interest:** The authors declare no conflict of interest.

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
