# Peer review of "Venture Capital and Chinese Firms’ Technological Innovation Capability: Effective Evaluation and Mechanism Verification"

_sustainability, doi:10.3390/su141610259_

Round 1

Reviewer 1 Report

This paper studies the impact of VC on technological innovation for Chinese listed companies. The conclusions are that VC has a positive influence through internal and external effects, but this impact depends on the size, industry, and type of organization. It recommends the government intervene in the factor market to favor innovation.

Although this paper is in line with special issue on Insights on Venture Capital and Sustainable Development of Enterprise, it does not sufficiently discuss the implications of this work on sustainability issues. Although a potential relationship between technical innovations (TI) improvements and corporate sustainability is evident, as noted in the introduction, the paper should develop a more detailed discussion on the sustainability aspects of its proposals.

Although the conclusions are similar to other studies done in other countries, the specific case of China and the consideration of listed companies provide novelty to this study. There are some novelty on  the explicit treatment of external mechanisms of influence of the VC on the TI. 

This paper process its data with regression techniques, and this study's robustness with PSM and other standard methods,  making the econometric approach  quite reasonable.

However,  paper's quality needs substantial improvement:

1)  English language has many orthographic mistakes and does not read well. The style of mathematical formulas is very poor, with inconsistent mathematical language -e.g. utility formula-. Other equations include  Chinese characters. Section separation and general appereance needs careful improvement as well.

2) Some arguments are unclear: e.g., the section on the theoretical mechanisms to arrive at the hypothesis proposal may be interesting, but it is not understood.

2) There are gaps: on page 5, equation 2 refers to equation 1, which does not appear in the text.

3) SA index is introduced in page 6 but not explained until page 11. It is claimed to use the DID method but not explicitly mentioned when/if used.

As a final consideration, I think that this paper deserves to be included in this special issue if the authors undergo major modifications as indicated above.

Author Response

Dear Reviewer:

Thank you for the comments concerning our manuscript entitled “Venture Capital and Chinese Firms’ Technological Innovation Capability: Effective Evaluation and Mechanism Verification” .Those comments are all valuable and very helpful for improving our paper, as well as the important guiding significance to our researches. We have studied comments carefully and have made revisions, which we hope will meet with approval. Revised portion are marked in red in the paper.

Please see the attached file for our detailed response to your comments.

Thanks.

Reviewer 2 Report

The paper analyses an interesting topic, and I like the research idea. The literature review puts the paper into the context quite clearly. I have two concerns about the empirical analysis:

- the paper focuses on A-shares, which are mainly available to Chinese investors, but not to foreign ones (although things are changing in recent years). How may this impact on the results? This aspect should be clarified, or at least mentioned as a potential limitation

- the paper focuses on the impact of VC on increasing patents obtained by companies. However, there may be an endogeneity problem that is not properly controlled for in the analysis. What if VCs are attracted by companies who already have a significant number of patents, on which they build to further expand their research and obtain further patents?

Author Response

(The authors gave the same response as above.)

Reviewer 3 Report

1)      Authors should add motivation and contribution in the introduction section

2)      Author should make one table in related study with existing work and their limitations?

3)      How the proposed model is beneficial in decentralized system?

4)      All the table must be improved and the text within the table must be aligned properly.

5)      The grammar and typos error must be taken care 

6)      Author should add advantages and disadvantages of the proposed model.

7)      Author should see the below manuscript and make suitability that how the proposed system model is ok with decentralized structure sustanibility?

a.       Kumar, R., & Tripathi, R. (2020). Secure healthcare framework using blockchain and public key cryptography. In Blockchain Cybersecurity, Trust and Privacy (pp. 185-202). Springer, Cham.

Author Response

(The authors gave the same response as above.)

Round 2

Reviewer 1 Report

Major concerns have adequately been addressed. 

Author Response

Dear Reviewer:

  Thank you for your endorsement and the comment concerning our manuscript entitled “Venture Capital and Chinese Firms’ Technological Innovation Capability: Effective Evaluation and Mechanism Verification”.  We have given the manuscript another proofreading addressing minor language issues (style and spelling). Please see the attached revised manuscript.  We hope that the revised manuscript will meet your expectation. 
   Best regards, 

Reviewer 2 Report

The authors did not really answer to the issues highlighted in the previous round:

- sample of A-shares: how does this affect the results, if at all? Or, at least, authors should explain why they use A-shares and what they are compared to other shares traded in Chinese stock markets;

- is there an endogeneity issue, due to the fact that VCs might be attracted by companies who already own patents before VC investments in the company?

Then, the authors themselves highlight another potential issue: VC funds are not uniformly spread across China, but concentrated in specific regions. Then, one should first explain why this happens, then if and how this may impact the results, and finally duly consider how to obtain robust findings despite this situation.

Author Response

Dear Reviewer:

Thank you for your comments concerning our manuscript entitled “Venture Capital and Chinese Firms’ Technological Innovation Capability: Effective Evaluation and Mechanism Verification” .Those comments are all valuable and very helpful for revising and improving our paper.

We have studied the comments carefully and have made revisions which we hope will meet your expectation. Please see our response to your comments and revised manuscript attached. Note that all revisions are highlighted in red in the paper.

Best regards,

Reviewer 3 Report

The manuscript can be accepted in the same format. All the comments are incorporated sucessfully.

Author Response

Dear Reviewer:

  Thank you so much for your endorsement concerning our manuscript entitled “Venture Capital and Chinese Firms’ Technological Innovation Capability: Effective Evaluation and Mechanism Verification”.  
   Best regards,